# What is known from the existing literature about self-management of pessaries for pelvic organ prolapse? A scoping review

Lucy Dwyer  ,[1,2] Dawn Dowding,[2] R Kearney[1,3]

[1]The Warrell Unit, Saint Mary's Hospital, Manchester University NHS Foundation Trust, Manchester Academic Health Science Centre, Manchester, UK
[2]Division of Nursing, Midwifery and Social Work, The University of Manchester School of Health Sciences, Manchester, UK
[3]Institute of Human Development, Faculty of Medical & Human Sciences, University of Manchester, Manchester, UK

**Correspondence to**
Lucy Dwyer;
lucy.dwyer@postgrad.manchester.ac.uk

## ABSTRACT

**Objectives** Pelvic organ prolapse can be managed with a pessary. However, regular follow-up may deter women due to the inconvenience of frequent appointments, as well as preventing autonomous decision making. Pessary self-management may be a solution to these issues. However, there remains a number of uncertainties regarding pessary self-management. This scoping review aims to map available evidence about pessary self-management to identify knowledge gaps providing the basis for future research.

**Design** Scoping review as detailed in the review protocol.

**Data sources** A search of MEDLINE, CINAHL, EMBASE and PsycINFO databases and a handsearch were undertaken during May 2021 to identify relevant articles using the search terms 'pessary' and 'self-management' or 'self-care'.

**Data extraction and synthesis** Data relevant to pessary self-management was extracted and the Mixed Methods Appraisal Tool used to assess empirical rigour. Thematic analysis was performed to evaluate the results.

**Results** The database search identified 82 publications. After duplicates and articles not meeting the inclusion and exclusion criteria were removed, there were 23 eligible articles. A hand search revealed a further 19 articles, resulting in a total of 42 publications.
Findings relevant to pessary self-management were extracted and analysed for the emergence of themes. Recurrent themes in the literature were; the characteristics of self-managing women; pessary care; factors associated with decision making about self-management; teaching self-management and cost benefit.

**Conclusions** Pessary self-management may offer benefits to some women without increased risk. Some women do not feel willing or able to self-manage their pessary. However, increased support may help women overcome this. Further in-depth exploration of factors which affect women's willingness to self-manage their pessary is indicated to ensure better understanding and support as available for other conditions.

## INTRODUCTION

Mechanical pessaries can effectively reduce symptoms of pelvic organ prolapse and therefore offer an effective alternative to surgery.[1] However, the regular pessary follow-up

required deters some women from this management option.[2–4] It has been suggested pessary self-management may reduce the frequency of pessary follow-up required. Moreover, it offers women the opportunity to make autonomous decisions about when and how to use their pessary. Nevertheless, there remains a number of uncertainties about pessary self-management including the potential benefits and risks and requirements of self-management teaching and follow-up care.[5] This scoping review aims to map available evidence about pessary self-management to identify knowledge gaps[6] providing the basis for future research.

## STRENGTHS AND LIMITATIONS OF THIS STUDY

⇒ The scoping review was undertaken following systematic and rigorous established guidelines to ensure transparency and reproducibility.
⇒ Critical appraisal of the identified publications enabled identification of methodological limitations within this subject area.
⇒ Studies not published in English were excluded which increases the risk of cultural bias in the findings.

## METHODS

As detailed in the published protocol (Dwyer *et al*, accepted by BMJ Open-Dec 21), the scoping review was conducted utilising the Joanna Briggs Institute scoping review methodology[7] and reported in accordance with Preferred Reporting Items for Systematic Reviews and Meta-Analyses extension for Scoping Reviews guidelines[8] (online supplemental material 1). The review was registered with The Open Science Framework (DOI 10.17605/OSF.IO/DNGCP). Between 5 May 2021 and 7 May 2021, a search of MEDLINE, CINAHL, EMBASE and PsycINFO was undertaken to identify relevant articles which met

## Inclusion criteria

- Original research
- Pessary for POP
- Published in English language
- Focuses on self-management of pessary for POP

## Exclusion criteria

- Not relevant to subject area
- Not published in English language
- Not original research including case reports

**Figure 1** Inclusion criteria. POP, pelvic organ prolapse.

the eligibility criteria (figure 1) using the search terms 'pessary' and 'self-management' or 'self-care' (see online supplemental materials 2–5). Handsearches for relevant and eligible publications not identified during the search were undertaken throughout May 2021.

A handsearch of the reference list of non-original research identified during the search but excluded, was conducted for additional publications which met the inclusion and exclusion criteria. Data relevant to the topic of pessary self-management were extracted and critical appraisal of all included publications undertaken using the Mixed Methods Appraisal Tool (MMAT).[9]

## PATIENT AND PUBLIC INVOLVEMENT

Members of the public and pessary users have not directly been involved with development of the review protocol or process. However, the need for research exploring pessary self-management was highlighted by The James Lind Alliance (JLA) Priority Setting Partnership for pessary and prolapse.[10] Several women with experience of pessaries participated in this partnership either as members of the steering group, by attending the consensus workshop or completing questionnaires. Understanding more about self-management was ranked third out of 20 priorities by the JLA Priority Setting Partnership.[10] The topic of the scoping review has therefore previously been identified and prioritised by patients and members of the public.

## RESULTS

The database search identified 82 publications. After duplicates and articles not accessible in the English language were removed, there were 64 remaining publications. After reviewing the identified articles in accordance with the inclusion and exclusion criteria (figure 1), there were 23 eligible articles. A hand search of reference lists from excluded non-original research papers, revealed a further 19 articles, resulting in a total of 42 publications (figure 2, online supplemental material 6). Ten of the 42 publications included were solely available as an abstract

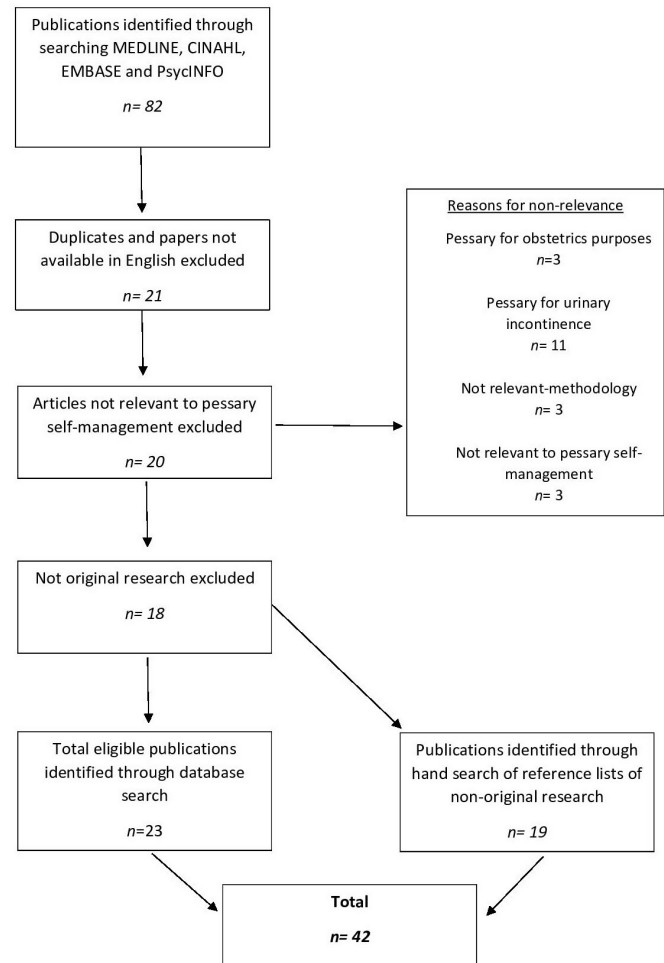

**Figure 2** Review flow chart.

which prevented thorough understanding and appraisal of both the findings and methodology.

As demonstrated in table 1, 42 studies included in the review were conducted across all six populated continents. Four (10%) took place in the UK. Half of the eligible studies took place in the USA. Only six (14%) of the eligible studies included in the review were interventional and two of these were secondary analysis of another included publication. A significant majority of the included studies were observational, most being conducted via case note review or questionnaires. A number of included publications report findings from the same sample of women as highlighted in online supplemental material 6. These findings have been reported separately due to the different aims of each publication.

Critical appraisal of the evidence using the MMAT (figure 3) reveals a number of recurring methodological limitations across the evidence base. First, due to the extent of retrospective case note reviews included, the quality of measurements is unlikely to be good due to the lack of a standardised protocol to record details of the pessary fitting and subsequent follow-up.[11] A further limitation is data completeness, due to both high levels of attrition within some of the randomised controlled trials included and a low response rate in many of the

**Table 1** Summary of included studies

| Total number of publications included | 42 | |
| --- | --- | --- |
| Continent where research based | | |
| Africa | 1 | |
| Asia | 7 | |
| Australasia | 4 | |
| Europe | 7 | |
| North America | 21 | |
| South America | 1 | |
| International | 1 | |
| Publication date—median (range) | 2015 | 1993–2021 |
| Publication type | | |
| Abstract | 10 | |
| Journal article | 32 | |
| Study design | | |
| Audit | 2 | |
| Case note review | 14 | |
| Cross sectional | 1 | |
| Laboratory study | 1 | |
| Observational | 17 | |
| Pre–post test | 1 | |
| Qualitative | 1 | |
| Quality improvement evaluation | 1 | |
| Randomised controlled trial (RCT) | 2 | |
| Secondary analysis of RCT | 2 | |

quantitative descriptive studies. This may introduce bias to the findings if those who are the most or least satisfied with pessary management are more or less inclined to respond or continue with study participation.[11] Only a small number of the quantitative non-randomised studies included, accounted for confounding variables within their findings by performing stratification, multivariate analysis or matching.[11] In the majority of studies that did not account for confounding variables the validity of the findings is questionable as various factors such as age, hormonal replacement status and prolapse stage could all have influenced the results. A further recurring limitation identified while appraising quantitative descriptive study was the lack of representativeness among study participants, whether pessary using women or healthcare professionals. Many studies used convenience samples whether only including pessary users who attended follow-up or healthcare professionals who were members of specialist organisations. Therefore, this may limit the generalisability of study findings.[11] Three of the included publications were surveys conducted among pessary practitioners.[12–14] It is acknowledged that data collected through self-reporting has a high risk of bias whether due to recall or social desirability bias.[15] However, these publications offer valuable insight into how healthcare

professionals perceive they are or should be, delivering pessary care, therefore, they have been included for this reason.

Findings relevant to pessary self-management were extracted from the included publications and analysed for the emergence of themes. Recurrent themes in the literature include the characteristics of self-managing women; pessary care; factors associated with decision making about self-management; teaching self-management and cost benefit.

### Characteristics of self-managing women

The extent of pessary self-management is difficult to gauge, though it appears to vary greatly throughout the world. Estimates vary from 18% of UK pessary practitioners to 53% of pessary practitioners in the USA offering pessary self-management.[13 14] Reasons for this variation are unclear but could be due to differences in care provision possibly due to associated costs of care. It is therefore possible those living in a country with high healthcare costs, may be more willing to self-manage to reduce the costs of attending follow-up appointments.

Several characteristics were associated with women being more likely to self-manage, including being younger,[16–23] being premenopausal,[23] being sexually active,[18 19 21] having a higher level of education,[18] prolonged pessary use,[24] having fewer comorbidities,[17–19 22 23] a lower stage of prolapse (one or two),[19 23] a smaller genital hiatus[22] and being diagnosed with atrophy.[21]

Age differences between women who self-manage or receive clinician-led care may be due to generational attitudes to pessary use or touching one's genitals, or the value associated with perceiving a clinic appointment as a social event with associated psychological and emotional support as proposed by Storey, Aston.[16] Conversely, older women or those with comorbidities may be less likely to self-manage due to a perception of, or actual increased difficulty. However, this has not yet been established.

Space occupying pessaries fill the vaginal cavity and therefore must be removed prior to sexual penetration.[25] Many other pessaries do not need to be removed for penetrative sex[26] However, 70% of women removed their pessary usually or always before sexual activity because their partner could feel the pessary, or due to discomfort.[27] Over half of the women removed the pessary before sexual intercourse after being asked by their partner rather than due to concerns about vaginal discharge or odour.[27] Despite daily pessary removal being associated with sexual activity, this was not correlated with sexual function.[27] Moreover, for women who were not sexually active, frequent pessary removal was associated with an improved relationship with one's partner.

There is disagreement in the literature regarding whether self-management affects the rate of women who choose to continue with pessary management of their prolapse. Self-management has been associated with pessary management continuation.[24 28–30] Conversely, other studies reported pessary self-management did not

| | 1. Qualitative studies | | | | | 2. Quantitative randomised controlled trial | | | | | 3. Quantitative non-randomized studies | | | | | 4. Quantitative descriptive studies | | | | |
|---|---|---|---|---|---|---|---|---|---|---|---|---|---|---|---|---|---|---|---|---|
| | 1.1-Appropriate approach | 1.2-Adequate data collection methods | 1.3 Are findings adequately derived? | 1.4- Interpretation sufficiently substantiated | 1.5-Coherence | 2.1-Appropriate randomisation | 2.2- Comparable groups at baseline | 2.3- Outcome data completeness | 2.4-Blinding | 2.5-Adherence to randomisation | 3.1-Representativeness | 3.2-Measurements appropriate | 3.3-Outcome data completeness | 3.4-Confounders accounted for | 3.5-Exposure as intended | 4.1-Relevant sampling strategy | 4.2- Representativeness | 4.3- Measurements appropriate | 4.4-Nonresponse bias | 4.5-Appropriate statistical analysis |
| Abdool et al, 2011 | | | | | | | | | | | 1 | 1 | 1 | 1 | 1 | | | | | |
| Bai et al, 2005 | | | | | | | | | | | | | | | | 1 | 0 | 0 | 1 | 1 |
| Brown et al, 2016 | | | | | | | | | | | | | | | | 1 | 0 | 1 | 0 | 1 |
| Bugge et al, 2013 | | | | | | | | | | | | | | | | 1 | 0 | 1 | 0 | 1 |
| Chan et al, 2019 | | | | | | | | | | | 1 | 1 | 0 | 1 | 1 | | | | | |
| *Chen et al, 2020a and Chen et al, 2020b** | | | | | | | | | | | 1 | 0 | 1 | 1 | 1 | | | | | |
| Chien et al, 2020 | | | | | | | | | | | 0 | 0 | 0 | 0 | 1 | | | | | |
| Clemons et al. 2004a, Clemons et al. 2004b and Clemons et al. 2004c** | | | | | | | | | | | 1 | 1 | 1 | 1 | 1 | | | | | |
| Cundiff et al, 2000 | | | | | | | | | | | | | | | | 1 | 1 | 1 | 0 | 1 |
| *Daneel et al, 2016 | | | | | | | | | | | | | | | | 1 | 1 | 0 | 1 | 0 |
| Goh et al, 2020 | | | | | | | | | | | | | | | | 1 | 0 | 1 | 1 | 1 |
| Hanson et al, 2006 | | | | | | | | | | | 1 | 0 | 1 | 0 | 1 | | | | | |
| Holubyeva et al, 2021 | | | | | | | | | | | 0 | 0 | 1 | 0 | 1 | | | | | |
| *Hooper et al, 2018 | | | | | | | | | | | 1 | 1 | 1 | 0 | 1 | | | | | |
| *Ibrahim et al, 2015 | | | | | | | | | | | | | | | | 1 | 1 | 1 | 0 | 1 |
| *Jacobs and Banks, 2010 | | | | | | | | | | | | | | | | 1 | 1 | 1 | 0 | 1 |
| Kearney and Brown, 2014 | | | | | | | | | | | 1 | 0 | 1 | 0 | 1 | | | | | |
| Khaja and Freeman, 2014 | | | | | | | | | | | | | | | | 1 | 0 | 1 | 0 | 1 |
| *Lammers et al, 2019 | | | | | | | | | | | 1 | 0 | 1 | 0 | 1 | | | | | |
| Kuhn et al, 2008 | | | | | | | | | | | | | | | | 1 | 1 | 1 | 1 | 1 |
| Ma et al, 2001 | | | | | | | | | | | 1 | 1 | 1 | 1 | 1 | | | | | |
| Manchana, 2011 | | | | | | | | | | | 1 | 1 | 1 | 0 | 1 | | | | | |
| Manonai et al, 2018 | | | | | | | | | | | 1 | 0 | 0 | 0 | 1 | | | | | |
| Meriwether et al, 2015a, Meriwether et al, 2015b and Fregosi et al, 2018** | | | | | | 1 | 1 | 0 | 1 | 1 | | | | | | | | | | |
| *Morcuende et al, 2018 | | | | | | | | | | | 1 | 0 | 1 | 0 | 0 | | | | | |
| Murray et al, 2017 | | | | | | | | | | | 1 | 1 | 1 | 0 | 1 | | | | | |
| Nemeth et al, 2013 | | | | | | | | | | | | | | | | 1 | 1 | 1 | 1 | 1 |
| *Pizarro-Berdichevsky et al, 2016 | | | | | | | | | | | 1 | 1 | 1 | 1 | 1 | | | | | |
| Ramsay et al, 2011 | | | | | | | | | | | 1 | 0 | 0 | 1 | 1 | | | | | |
| Sarma et al, 2009 | | | | | | | | | | | | | | | | 1 | 1 | 0 | 0 | 1 |
| Storey et al, 2009 | 1 | 1 | 1 | 1 | 1 | | | | | | | | | | | | | | | |
| Sulak et al, 1993 | | | | | | | | | | | | | | | | 0 | 0 | 0 | 1 | 1 |
| Tam et al, 2019 | | | | | | 1 | 1 | 1 | 1 | 1 | | | | | | | | | | |
| Thys et al, 2020 | | | | | | | | | | | 1 | 1 | 0 | 0 | 1 | | | | | |
| Tenfelde et al, 2015 | | | | | | | | | | | | | | | | 1 | 1 | 1 | 1 | 1 |
| Wu et al, 1997 | | | | | | | | | | | | | | | | 1 | 1 | 1 | 1 | 1 |
| Yoshimura et al, 2020 | | | | | | | | | | | 1 | 1 | 1 | 0 | 0 | | | | | |

\* Abstract

\*\* Clustered studies where secondary analysis of the same sample of women was performed

http://mixedmethodsappraisaltoolpublic.pbworks.com/w/file/fetch/140056890/Reporting%20the%20results%20of%20the%20MMAT.pdf

**Figure 3** MMAT results. MMAT, Mixed Methods Appraisal Tool.

affect pessary continuation rates when compared with either nurse or doctor clinician-led care at 1 year.[19 31 32] It is important to note the studies where no relationship was established were based on shorter-term data. Therefore, it is possible this does not allow sufficient time to measure a difference.

## Pessary care

There is a wide range of pessaries available to manage pelvic organ prolapse and these are typically classed as either a support or space occupying pessary.[25] Frequently used support pessaries in the UK include ring pessaries, rings with support and shaatz pessaries.[13] There is agreement among the literature that support pessaries are suitable for women to self-manage, with ring pessaries being 'acceptable'[12] and 'easy'[33–35] to remove and insert. Wu et al[33] suggest women may feel comfortable performing self-management with a ring pessary as it is a similar size and shape to a diaphragm. Whether there is any relationship between willingness or ability to self-manage a pessary, and previous use of a vaginal device has

not previously been explored and research into this may provide important insight. Kearney and Brown[2] report more women were willing or able to self-manage with a ring than sieve pessary. Further research to assess whether this is replicated among a larger population of women is indicated.

Commonly used space occupying pessaries include shelf, gell-horn and cube pessaries.[13 25] These work by providing both structural support and filling the vaginal cavity to prevent vaginal descent.[25] Women using space occupying pessaries are less likely to perform self-management than those using support pessaries.[19 23] Moreover, Clemons et al[36] state gell-horn pessaries are too difficult to self-manage, a perception shared by many pessary practitioners.[12 14] Despite this widespread belief, a number of studies suggest self-management of space occupying pessaries such as shelf or gell-horn pessaries is possible,[37 38] if in some cases more difficult.[39] After self-management teaching, 62% and 47% of women continued to self-manage their gell-horn pessary after 1 and 5 years, respectively.[38] This compares similarly to the percentages of women self-managing with other types of pessaries. Bai et al[34] state the donut pessary is difficult for a woman to remove independently, however, no rigorous evidence to support this statement is provided. More perplexing is Hanson et al's[39] assertion that cube pessaries are among the most difficult pessary to remove. Again, there is no rigorous evidence regarding this statement. Furthermore, according to the manufacturers, it is a requirement to self-manage a cube pessary as it should be removed on a nightly basis.[40] This is due to the increased risk of complications due to the greater surface area covered by the cube pessary and the mild suction.[41]

At present, there is a lack of evidence about the required frequency of pessary removal by self-managing women to inform practice. The frequency of pessary removal and insertion will vary in accordance with the reasons for self-management. The most frequently reported frequency of pessary removal and insertion was daily to a couple of times a week.[17–19 32] A number of protocols provided specific advice regarding frequency of pessary removal. Whether removal each time they bathed[38] or removal and to leave the pessary out overnight once or twice a week.[30 42] Seventy per cent (n=35) of women followed this advice given and removed the pessary at least weekly.[42] One self-managing woman left the gell-horn pessary in situ for 6 months without removal and experienced no issues. This is unsurprising, as for most UK centres, standard clinician-led care is to follow-up pessaries on a 6 monthly basis.[43] On this basis, Kearney and Brown[2] recommended self-managing women remove their pessary at a minimum of 6 monthly. Chan et al[32] advised women not to leave the pessary in situ for longer than 3 months, which mirrored clinician-led care at that organisation. A self-management protocol advising women to remove their pessary on a weekly basis was cited as the reason for pessary discontinuation as women found it too burdensome.[35] While there is no evidence to inform guidance about recommended

frequency of pessary removal and insertion, instructing women to remove their pessary very frequently may deter women from self-management but also reduce the autonomy self-management offers.

The majority of studies suggest pessary self-management offers benefits in terms of women's comfort, convenience, perceived access to help and support and feeling of independence.[2 39] Self-management reduces complications when compared with clinician-led care, with an overall complication rate for self-managing women of 10%–16%,[35 37 44] compared with complication rates of 62% for women receiving clinician-led care.[35] Studies report a lower rate of vaginal erosions for self-managing women.[19 23] However, no difference was reported between women self-managing or receiving clinician-led care regarding vaginal pain, discharge or irritation.[21] Vaginal pain or discomfort was experienced by 40% (n=37) of women during removal or insertion of their gell-horn pessary.[38] As there was no comparison with women receiving clinician-led care it is not known whether the pain and discomfort reported by women is comparable to that experienced by women during clinician-led pessary changes.

There is mixed evidence for the effect on vaginal discharge, with one study reporting an increase for one woman who commenced self-management[2] and another an increase in discharge for women who left their pessary in situ continuously.[20] Despite identifiable differences in the vaginal microenvironment associated with frequency of removal, this did not appear to be clinically significant with regards to women's symptoms after 3 months.[45] It is possible if conducted over a longer period, such as 6 months which is more akin to the typical length of time a pessary is left in situ,[43] greater differences might have been identified. Furthermore, due to the frequency of removal the women were grouped into, the findings cannot be generalised to the large population of women who do not remove their pessary at all.

## Factors associated with decision making about self-management

In publications which explored willingness to self-manage a pessary, the percentage of willing women varied significantly, from 3% to 83%.[16 22 23 32 36 46–49] Reports of women's levels of ability to self-manage their pessary also differs extensively between different studies, ranging between 40% and 86%.[2 21 24 28 35 37 41 44 50] Furthermore, ability to self-manage a pessary does not appear to be a straightforward concept. Twenty per cent of women reported occasional difficulty removing their pessary or required assistance from a healthcare professional.[42] After teaching, 17% of women initially found self-management difficult, however, this reduced significantly with repeated attempts.[51] It is unclear why self-management ability may fluctuate. Manual dexterity was cited as an important aspect of self-management ability.[52] Kearney and Brown[2] found issues with this to be the most common barrier,

causing seven percent of women to discontinue self-management after 6 months.

Three studies explored the reasons why women self-managed their pessary. Mostly women removed or readjusted their pessary when it was slipping or uncomfortable,[34] or because they desired flexibility in how and when they used their pessary.[16] After 6 months of self-management the most frequently cited reason for pessary removal was removal and reinsertion at 6 months as advised by their clinician (36%, n=16). This was followed by removal for cleaning (31%, n=14), removal to aid defecation (11%, n=5), removal due to discomfort (22%, n=10), removal to have sex (13%, n=6) removal for a smear or other procedure (4%, n=2) and removal before a holiday (2%, n=1). It is possible the reasons for removal may be different if collected over a longer period than 6 months. However, it offers clinicians some understanding into self-management behaviours of women.

Women who experienced pessary-related complications were also less likely to opt for self-management, or to discontinue self-management whether due to patient request[37] experiencing unexpected bleeding, or changing to a pessary not suitable for self-management.[2] Physical barriers were the most cited reason women gave to Kearney and Brown[2] for non-participation in the study. However, further detail was not provided therefore it is unclear exactly what these physical barriers were. Another physical barrier to self-management reported by 6% of women was feeling 'too old'. Whether age as a barrier relates to physical ability, a mind-set or a combination of the two is not clear.

A reason many women gave for not wanting to self-manage was a lack of confidence.[16 53] However, despite a lack of confidence undertaking the self-management role, the same women demonstrated high levels of confidence addressing daily challenges with their pessary, such as readjusting it as required.[16] It may also be women feel more able to access support if receiving clinician-led care. Conversely, women who commenced pessary self-management reported better access support than those receiving clinician-led care.[2] The intimate nature of pessary care is also a consideration, some women reported feeling uncomfortable touching their genitalia.[16] This was also identified by Kearney and Brown[2] where 11% of women declined self-management due to the nature of pessary care. A similar issue reported by Abdool et al[54] was many women expressing a preference not to handle a pessary. Thirteen per cent of women declined self-management, reporting clinicians found removal or insertion difficult and therefore they did not feel able to self-manage the pessary independently.[2]

### Teaching self-management

At present, there are no evidence-based guidelines to specify the requirements of self-management teaching.[55] The identified literature provides some evidence regarding the included constituents of self-management teaching. Where reported, self-management teaching took between 45 min and an hour.[2 56] The level of detail provided about self-management teaching varied greatly. Therefore, it was not always clear whether details of basic instructions were simply a brief description, or an accurate representation of the teaching provided. In all identified studies which provided details about the teaching, women were given instructions on removal and insertion of the pessary.[2 29 51 56] Kearney and Brown[2] specified that women were supervised practicing a pessary change, however, this was not mentioned elsewhere.

Kearney and Brown[2] state that as well as face-to-face training, women were provided with written information and access to an online video. Both were designed using feedback from pessary users who attended a focus group. Women who received written information were more satisfied, more confident and demonstrated increased knowledge regarding every aspect of pessary care covered by the brochure when compared with women solely given verbal information.[48]

Within the identified literature women were taught to self-manage by specialist nurses[37 56] or specialist women's health physiotherapist.[2] Kearney and Brown[2] originally intended to have a specialist nurse undertake this role. However, they believe having a physiotherapist who would typically perform a rehabilitative role rather than delivering medical interventions was beneficial to the service evaluation. Only one study evaluated outcomes following self-management teaching[51] found after only one tutorial, new pessary using women had increased confidence to self-manage their pessary and described it as 'easy'.

### Cost benefit

While the primary goal of pessary self-management is an improved experience for women, there is also the potential for cost savings if the number of follow-up appointments required can be reduced both for the woman and the service.[29 51] While the cost of attending frequent follow-up may be a consideration for women attending appointments anywhere, it is important to clarify the women audited by Goh et al[51] lived in a 'poor' rural farming area of Uganda, one of the poorest countries in the world.[57] Therefore, the impact of costs to travel to an appointment are likely to be more of a consideration or deterrent, for women in this audit compared with the general population. Detailed cost analysis of self-management versus clinician-led care suggests potential for annual cost savings of £5500 for the Trust and £8400 for commissioners for 50 women.[2] This calculation is based on 90 min of a band six nurse or physiotherapist's time to provide initial self-management teaching and subsequent telephone follow-up. It is important to note, the significant cost savings calculated are based on clinician-led pessary care being provided by doctors, whereas at most organisations in the UK it is provided by a combination of doctors and nurses.[58] This may affect projected cost savings.

## DISCUSSION

Despite a number of factors being associated with likelihood to self-manage,[16–24] a causative relationship has not been established with any. As physical ability does not appear to affect willingness to self-manage a pessary,[42 51] further exploration is required to enable us to understand why such a large number of women are unwilling to undertake this role.[16 22 23 32 33 36 46–49]

As suggested in the literature, it is unsurprising with increasing practice and confidence, self-management would get easier.[51] It may also be due to pessary-related factors such as the position of the pessary changing and becoming harder to reach or more immovable. Alternatively, it might be that dynamic characteristics of the woman such as hormonal changes affecting the flexibility or lack of, of the surrounding vaginal tissues, physical flexibility and dexterity fluctuations cause occasional difficulty. Due to the fluctuating nature of self-management ability,[42] it is questionable whether a decision can be made on one assessment. Despite this, there is no robust evidence regarding the skills and ability required to self-manage a pessary, therefore, further research in this area is indicated.

The finding of the scoping review demonstrate the lack of evidence to guide decision making regarding pessary self-management. However, there is potential for the weight of evidence to appear greater than it is due to repetition. Ten of the included studies had been solely published in abstract format despite many of them being presented a number of years ago.[19 22 24 28 37 44 47 50 53 59] The decision was made to include these abstracts despite not being published in full to ensure very recent data was included. However, it has highlighted the potential for publication bias in the literature related to self-management.

Despite critical appraisal of evidence not being a traditional component of scoping reviews,[7] assessment of the quality of the included papers has enabled us to identify recurring themes in the methodological limitations of the evidence base. This highlights the need for well-designed research studies, particularly randomised controlled trials within this field to improve rigour within the literature. A further limitation is restriction to the four databases searched, which may result in failure to identify eligible, relevant publications. The search terms utilised included 'self-management' and 'self-care'. However, it is possible not all centres use these terms to describe teaching women to remove and insert their pessaries. In which case only instances where authors use the terms self-management or self-care or perceive self-management of pessaries as something other than standard care will have been identified.

In conclusion, despite the lack of rigorous evidence, the publications identified in this scoping review suggest pessary self-management offers benefits to women[2 39] with no increased risk of complications.[19 23 35 37 44] It is clear some women do not feel willing or able to self-manage their pessary.[2 16 21–24 28 32 33 35–37 41 44 46–50] Therefore, in-depth exploration of women's beliefs and attitude towards pessary self-management is required to further our understanding of the barriers women experience and to determine whether increased support to help women overcome these issues or concerns may address this. Further rigorous evidence is also required to clarify whether or not clinician-led follow-up is necessary for self-managing women, or, whether self-managing women could instead request an appointment only if they experience problems or have pessary-related concerns. In a cohort of women receiving clinician-led care, reliance on patient-reported symptoms to identify pessary complications was found to miss 27% of women with an excoriation.[60] However, whether this would be the case in self-managing women who would be removing their pessary more frequently has not been established.

The findings of the TOPSY study[55] expected to be published in 2022 are anticipated to clarify certain aspects of self-management care. However, further in-depth exploration of factors which affect women's willingness to self-manage their pessary is indicated to ensure the issue is better understood and women are better supported to self-manage their prolapse as those with other conditions have been previously.

**Contributors** LD devised the scoping review question, methodology, performed the literature search, data extraction and drafted this manuscript. RK substantively contributed to the development of the scoping review question, methodology, performed data extraction and critical appraisal of a subsample of the included studies and has revised and approved this manuscript. DD substantively contributed to the development of the scoping review question, methodology and revised and approved this manuscript. LD accepts full responsibility for the work and conduct of the review, had full access to the data, and controlled the decision to publish.

**Funding** LD, Clinical Doctoral Research Fellow, NIHR300519 is funded by Health Education England (HEE)/National Institute for Health Research (NIHR) for this research project.

ORCID iD
Lucy Dwyer http://orcid.org/0000-0002-0284-873X

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
