## [Reviewer comments · BMJ Open]

ARTICLE DETAILS

TITLE (PROVISIONAL)	What is known from the existing literature about self-management of pessaries for pelvic organ prolapse? A scoping review.
AUTHORS	Dwyer, Lucy; Dowding, Dawn; Kearney, R

VERSION 1 – REVIEW

REVIEWER	Daskalakis, George University of Athens, Obstetrics and Gynecology
REVIEW RETURNED	12-Jan-2022

GENERAL COMMENTS	Dear editor, i thank you for providing me the opportunity to review this manuscript. Scoping reviews are expected to increase within the next years, and this article seems to be very nicely organized. The authors should ideally present the subdomains of the Mixed Methods Appraisal Tool as it is impossible for readers to understand its subsection. Ideally this info should be inserted in the captions section. The conclusion section should be transparent and the use of vague phrases should be avoided ex. "appears pessary self-management offers benefits to some women" (which women?); "Further research is indicated to clarify the follow-up required for self-managing women, as well as the constituents of effective self-management teaching." (in which direction the research should be focused?) Overall, while i do like the study, the discussion section is very narrow and does not provide concise information and i believe it would benefit from a revision in which the authors should follow the above recommendations
---

REVIEWER	Pizzoferrato, Anne-Cécile University Hospital Centre Caen, Obstetrics-Gynecology and Reproductive Medicine
REVIEW RETURNED	18-Feb-2022

GENERAL COMMENTS	Thanks to the editor for offering me to review this very interesting work. Self-management of pessaries in case of pelvic organ prolapse (POP) becomes an important issue when seeking to promote the use of the pessary (especially among younger women), over the long term and to postpone surgery for as long as possible.
--

	Several learned societies recommend pessary fitting as a first-line treatment option regardless of the type, the stage of pelvic organ prolapse or age of the patient. This device should not be offered only to older women. My comments : Introduction : well written Methods : Why did the authors include practice surveys? I wonder how these studies appear in the search with the search terms stated. The MMAT is a very interesting tool for critical appraisal of the included publications. Results : Potential bias are very well explained Section “Pessary care”: This section particularly focused about the use of the gellhorn pessary. Page 10 : Lines 30-31, a sentence bothers me : “Women using space occupying pessaries are less likely to perform self-management than those using support pessaries ». Cube pessaries witch are also space occupying pessaries are the most used pessaries in France, just after ring pessaries (Pizzoferrato et al., J Womens health 2021). They are mainly used in young women able to remove it every day, as indicated by the manufacturer. They can also be effective in case of urinary incontinence associated with POP. Lines 46-47 : “Hanson et al’s assertion” also makes me wonder... Lines 59-60 : the sentence : “The most frequently reported frequency of pessary removal and insertion was daily to a couple of times a week[15-17, 30]” was also about space occupying pessaries or about all types of pessaries ? Can the authors clarify? Page 11: The sentence “While there is no evidence to inform guidance about recommended frequency of pessary removal and insertion, instructing women to remove their pessary very frequently may deter women from self-management but also reduce the autonomy self-management offers » is very interesting. My opinion is that professionals should explain that women have the choice of the pessary removal frequency, especially when using ring pessaries: women can use their pessary all the time or only during activities at risk of pelvic floor dysfunctions, with more or less cleaning frequency depending on their activities (in particular sexuality, defecation, etc.). It is the possibility of managing their pessary at their convenience which makes it possible to maintain a pessary for a long time. But this remains to be assessed more specifically... Section “Factors associated with decision making about self-management” Page 12 : Lines 8-9 : Authors have to moderate the sentence “Many women refused to self-manage their pessary when offered this option, preferring to return regularly for clinician-led care instead[32, 45] ». It is important to closely supervise the women at the time of a pessary fitting and to promote self-management at each consultation because it is sometimes difficult from the first
--	--

	consultation. The objective should be to promote self-management of pessaries at each appointment. Page 13 : Line 3 : "A reason many women gave for not wanting to self-manage was a lack of confidence". You are right, but with appropriate supervision many women become independent after several consultations; particularly with the ring pessary which is fairly easy to use. Discussion : Page 14, lines 58-59 : "Due to the fluctuating nature of self-management ability[41], it is questionable whether a decision can be made upon one assessment." I totally agree. Tables and figures : The images are not numbered correctly (pages 22,23,24). Why the data extraction table was provided as supplementary material? It is however a very interesting table. To better analyze the table, the authors could specify the types of pessaries used in the studies (if the data is available). The type of pessary is, in my opinion, a determining factor in the possibility of self-management of a pessary. Gellhorn pessaries are not often used in France and are quite difficult to remove. I would also like to see a column with the stage of POP and the average age of the women included in the selected studies, as well as the average follow-up. To conclude, I would be very interested in the results of the TOPSY trial.
--	---

VERSION 1 – AUTHOR RESPONSE

The authors should ideally present the subdomains of the Mixed Methods Appraisal Tool as it is impossible for readers to understand its subsection. Ideally this info should be inserted in the captions section.

Actioned as advised

The conclusion section should be transparent and the use of vague phrases should be avoided ex. "appears pessary self-management offers benefits to some women" (which women?); "Further research is indicated to clarify the follow-up required for self-managing women, as well as the constituents of effective self-management teaching." (in which direction the research should be focused?)

Actioned

Methods : Why did the authors include practice surveys? I wonder how these studies appear in the search with the search terms stated.

Despite the limitations of practice surveys, Khaja and Freeman's publication was identified using the search terms specified during the search of Embase. As it met the eligibility criteria it was included in the study. The two other practice surveys were identified during hand searches of reference lists. They meet the eligibility criteria and therefore were included. In view of the reviewer's comments, the use of practice surveys and limitations of this data has been added to the critical appraisal section.

Page 10 : Lines 30-31, a sentence bothers me : "Women using space occupying pessaries are less likely to perform self-management than those using support pessaries ». Cube pessaries witch are also space occupying pessaries are the most used pessaries in France, just after ring pessaries (Pizzoferrato et al., J Womens health 2021). They are mainly used in young women able to remove it every day, as indicated by the manufacturer. They can also be effective in case of urinary incontinence associated with POP.

Actioned with clarification-thank you for pointing out that this was not clear

Lines 59-60 : the sentence : “The most frequently reported frequency of pessary removal and insertion was daily to a couple of times a week[15-17, 30]” was also about space occupying pessaries or about all types of pessaries ? Can the authors clarify?

This was all types of pessaries-this will now be clear to readers as I have added details of the types of pessaries used in each study as per the recommendation below.

Section “Factors associated with decision making about self-management”

Page 12 : Lines 8-9 : Authors have to moderate the sentence “Many women refused to self-manage their pessary when offered this option, preferring to return regularly for clinician-led care instead[32, 45]

Removed as this did not add anything additional to the following sentence where the percentages of willing women adds the required detail.

Tables and figures :

The images are not numbered correctly (pages 22,23,24).

Actioned-apologies for this oversight

To better analyse the table, the authors could specify the types of pessaries used in the studies (if the data is available). I would also like to see a column with the stage of POP and the average age of the women included in the selected studies, as well as the average follow-up.

Actioned

VERSION 2 – REVIEW

REVIEWER	Pizzoferrato, Anne-Cécile University Hospital Centre Caen, Obstetrics-Gynecology and Reproductive Medicine
REVIEW RETURNED	17-May-2022
GENERAL COMMENTS	I feel the authors have answered and addressed all my previous comments